# Enhanced Malignant Phenotypes of Glioblastoma Cells Surviving NPe6-Mediated Photodynamic Therapy are Regulated via ERK1/2 Activation

**DOI:** 10.3390/cancers12123641

**Published:** 2020-12-04

**Authors:** Tatsuya Kobayashi, Makoto Miyazaki, Nobuyoshi Sasaki, Shun Yamamuro, Eita Uchida, Daisuke Kawauchi, Masamichi Takahashi, Yohei Otsuka, Kosuke Kumagai, Satoru Takeuchi, Terushige Toyooka, Naoki Otani, Kojiro Wada, Yoshitaka Narita, Hideki Yamaguchi, Yoshihiro Muragaki, Takakazu Kawamata, Kentaro Mori, Koichi Ichimura, Arata Tomiyama

**Affiliations:** 1Division of Brain Tumor Translational Research, National Cancer Center Research Institute, 5-1-1 Tsukiji, Chuo-ku, Tokyo 104-0045, Japan; tatskoba@ncc.go.jp (T.K.); makmiyaz@ncc.go.jp (M.M.); nobsasaki@ks.kyorin-u.ac.jp (N.S.); yamamuro.shun@nihon-u.ac.jp (S.Y.); eita1985@saitama-med.ac.jp (E.U.); dkawauch@ncc.go.jp (D.K.); 2Department of Neurosurgery, National Defense Medical College, 3-2 Namiki, Tokorozawa, Saitama 359-8513, Japan; doc32007@ndmc.ac.jp (Y.O.); cln219@ndmc.ac.jp (K.K.); stake@ndmc.ac.jp (S.T.); toyo@ndmc.ac.jp (T.T.); stingray@ndmc.ac.jp (K.W.); kichimur@ncc.go.jp (K.I.); 3Department of Neurosurgery, Tokyo Women’s Medical University, 8-1 Kawada-cho, Shinjuku-ku, Tokyo 162-8666, Japan; ymuragaki@twmu.ac.jp (Y.M.); tkawamata@twmu.ac.jp (T.K.); 4Faculty of Advanced Techno-Surgery, Tokyo Women’s Medical University, 8-1 Kawada-cho, Shinjuku-ku, Tokyo 162-8666, Japan; 5Department of Cancer Cell Research, Sasaki Institute, Sasaki Foundation, 2-2 Kandasurugadai Chiyoda-ku, Tokyo 101-0062, Japan; h-yamaguchi@po.kyoundo.jp; 6Department of Neurosurgery, Kyorin University Faculty of Medicine, 6-20-2 Shinkawa, Mitaka, Tokyo 181-8611, Japan; 7Department of Neurological Surgery, Nihon University School of Medicine, 30-1 Ohyaguchi, Kamicho, Itabashi-Ku, Tokyo 173-8610, Japan; otani.naoki@nihon-u.ac.jp; 8Department of Neuro-Oncology/Neurosurgery, Saitama Medical University International Medical Center, 1397-1, Yamane, Hidaka-City, Saitama 350-1298, Japan; 9Department of Neurological Surgery, Chiba University Graduate School of Medicine, 1-8-1 Inohana, Chuo-ku, Chiba-shi, Chiba 260-8677, Japan; 10Department of Neurosurgery and Neuro-Oncology, National Cancer Center Hospital, 5-1-1 Tsukiji, Chuo-ku, Tokyo 104-0045, Japan; masataka@ncc.go.jp (M.T.); yonarita@ncc.go.jp (Y.N.); 11Department of Neurosurgery, Tokyo General Hospital, Tokyo 165-8906, Japan; kentaro.mori@mt.strins.or.jp

**Keywords:** photodynamic therapy, talaporfin, resistance, migration, ERK1/2, glioblastoma

## Abstract

**Simple Summary:**

The molecular machineries regulating resistance against photodynamic therapy (PDT) using talaporfin sodium (NPe6) (NPe6-PDT) in glioblastomas (GBM)s and mechanisms underlying the changes in GBM phenotypes following NPe6-PDT remain unknown. Herein, we established an in vitro NPe6-mediated PDT model using human GBM cell lines. NPe6-PDT induced both caspase-dependent and -independent GBM cell death in a NPe6 dose-dependent manner. Moreover, treatment with poly (ADP-ribose) polymerase inhibitor blocked NPe6-PDT-triggered caspase-independent GBM cell death. Next, it was revealed resistance to re-NPe6-PDT, migration, and invasion of GBM cells that survived following NPe6-PDT (NPe6-PDT-R cells) were enhanced. Immunoblotting of NPe6-PDT-R revealed that only ERK1/2 activation exhibited the same trend as migration. Importantly, treatment with the MEK1/2 inhibitor trametinib reversed resistance against re-NPe6-PDT and suppressed the enhanced migration and invasion of NPe6-PDT-R cells. Overall, enhanced ERK1/2 activation is suggested as a key regulator of elevated malignant phenotypes of GBM cells surviving NPe6-PDT.

**Abstract:**

To manage refractory and invasive glioblastomas (GBM)s, photodynamic therapy (PDT) using talaporfin sodium (NPe6) (NPe6-PDT) was recently approved in clinical practice. However, the molecular machineries regulating resistance against NPe6-PDT in GBMs and mechanisms underlying the changes in GBM phenotypes following NPe6-PDT remain unknown. Herein, we established an in vitro NPe6-mediated PDT model using human GBM cell lines. NPe6-PDT induced GBM cell death in a NPe6 dose-dependent manner. However, this NPe6-PDT-induced GBM cell death was not completely blocked by the pan-caspase inhibitor, suggesting NPe6-PDT induces both caspase-dependent and -independent cell death. Moreover, treatment with poly (ADP-ribose) polymerase inhibitor blocked NPe6-PDT-triggered caspase-independent GBM cell death. Next, it was also revealed resistance to re-NPe6-PDT of GBM cells and GBM stem cells survived following NPe6-PDT (NPe6-PDT-R cells), as well as migration and invasion of NPe6-PDT-R cells were enhanced. Immunoblotting of NPe6-PDT-R cells to assess the behavior of the proteins that are known to be stress-induced revealed that only ERK1/2 activation exhibited the same trend as migration. Importantly, treatment with the MEK1/2 inhibitor trametinib reversed resistance against re-NPe6-PDT and suppressed the enhanced migration and invasion of NPe6-PDT-R cells. Overall, enhanced ERK1/2 activation is suggested as a key regulator of elevated malignant phenotypes of GBM cells surviving NPe6-PDT and is therefore considered as a potential therapeutic target against GBM.

## 1. Introduction

Glioblastoma (GBM) is the most common malignant intracranial tumor arising from the brain parenchyma and has the poorest prognosis, with median survival of only 12–15 months despite standard multidisciplinary treatment with maximal surgical resection, followed by chemoradiotherapy [1]. This poor prognosis can be attributed to the limited surgical excision of GBM, due to its highly infiltrative growth into the normal brain tissue, need to minimize the damage to normal brain tissue due to surgery, and high resistance of this tumor to chemoradiotherapy. Numerous efforts have been put into improving GBM prognosis for decades, albeit without much success. Hence, novel therapies against GBM must be urgently developed.

Photodynamic therapy (PDT) was originally aimed at killing target cells with minimal invasiveness and toxicity [2,3,4]. Following the uptake of a photosensitizer by specific target cells, highly toxic free radicals are produced in these cells through light-induced excitation of the photosensitizer; these radicals induce cell death. Based on this principle, PDT-mediated specific killing of target tumor cells without damaging the surrounding normal tissue has been attempted and approved as a novel therapy against refractory tumors, including GBMs, in recent years [4,5]. In particular, in patients with GBM, intraoperative PDT using laser and talaporfin sodium (mono-l-aspartyl chlorin e6, NPe6) against residual tumors and postoperative PDT to tumor bed have been developed because of the highly infiltrative nature of GBMs, and positive outcomes have been reported when this treatment was added to clinical management of GBMs. In 2013, in a phase II clinical trial, patients with recurrent GBMs who underwent NPe6-mediated PDT (NPe6-PDT) showed a highly favorable response rate of 12 months for progression-free survival (PFS) and 24.8 months for overall survival (OS) (one-year survival rate, 100%) [6]. In addition, in a subsequent prospective observational study of newly diagnosed GBM, the PFS and OS of patients who received additional NPe6-PDT were significantly prolonged by 19.6 and 27.4 months, respectively, compared with those of patients who did not receive this treatment, with the prolongation of PFS being particularly noteworthy [7]. Unfortunately, however, the molecular mechanisms underlying resistance to PDT or the regulation of oncogenic phenotypes of GBM cells, such as resistance against NPe6-PDT or invasiveness, during relapse following NPe6-PDT are not well elucidated to date. Notably, with regard to the change in recurrence pattern, a previous study reported a tendency of GBMs to gain a more invasive phenotype in the group treated with NPe6-PDT than in the group treated without it [7]. Consequently, GBM cells that survive NPe6-PDT may gain a more aggressive phenotype than that before treatment. Based on these reports, the molecular mechanisms underlying resistance to NPe6-PDT and switch in tumorigenic phenotypes of GBM cells and their regulation during relapse following NPe6-PDT must be explored to further improve the outcomes of NPe6-PDT for GBMs.

The mitogen-activated protein kinase (MAPK) cascade, which is initiated by the activation of receptor tyrosine kinases (RTKs), followed by that of Ras-rapidly accelerated fibrosarcoma (RAF; A-Raf, B-Raf, and C-Raf), mitogen-activated protein kinase 1/2 (MEK1/2), and mitogen-activated protein kinase 3/1 [extracellular signal-regulated kinase1/2 (ERK1/2)], is one of the downstream signaling pathways of RTKs, which play pivotal roles in regulating cellular functions, such as migration, proliferation, or survival [8,9,10]. In various cancer systems, this cascade is frequently hyperactivated due to genetic alterations of the related molecules, thereby contributing to oncogenesis [8,9,10,11]. Hyperactivation of RTKs or functional downregulation of neurofibromin 1 (*NF1*)—a GTPase-activating protein that negatively regulates Ras—induced by genetic mutations contributes to the regulation of oncogenic properties of GBMs [11,12,13,14]. Consequently, the signaling molecules of this cascade, such as RAFs, MEKs, or ERKs, have been investigated as therapeutic targets for GBM [8,13].

To this end, in the present study, we explored the molecular mechanisms regulating the resistance of GBM cells to NPe6-PDT, as well as the switch in the phenotypes of GBM cells that survive NPe6-PDT and their regulation. Our results indicate that ERK1/2 activation is essential for regulating the resistance of GBM cells to NPe6-PDT. Furthermore, the migration and invasion abilities of cells that survived NPe6-PDT were enhanced, specifically within 14 days after laser irradiation, in an ERK1/2 activity-dependent manner; therefore, MAPK inhibition may be an effective strategy to enhance the efficacy of NPe6-PDT against GBMs.

## 2. Materials and Methods

### 2.1. Reagents and Antibodies

RPMI 1640 medium (#11875176), 10% fetal bovine serum (FBS), 1% penicillin–streptomycin (#15140122), propidium iodide (PI, #P4864), Hoechst 33342 (Ho, #B2261), Lipofectamine RNAiMAX, and Alexa 546 phalloidin (#A22283) were obtained from Thermo Fisher Scientific (Tokyo, Japan). The 5-FU was purchased from FUJIFILM Wako Pure Chemical Corporation (Osaka, Japan). PJ34 was obtained from Merck (Tokyo, Japan). (CellTiter-Glo 2.0 Assay (#G9241) kit was obtained from Promega (Tokyo, Japan). Signal Enhancer HIKARI (#02270-81) protease inhibitor cocktail (#25955-11) and phosphatase inhibitor cocktail EDTA-free (07575-51) were purchased from Nacalai Tesque Inc. (Kyoto, Japan). Western Lightning Plus-ECL (#NEL1015001EA) was purchased from PerkinElmer Japan (Yokohama, Japan). NPe6 was kindly provided by Meiji Seika Co., Ltd. (Tokyo, Japan).

Primary antibodies against apoptosis-inducing factor (AIF) (#D39D2), Bcl-2 homologous antagonist/killer (Bak, #12105), Bcl-2-associated X protein (Bax, D2E11), phospho-Akt (Ser 473) (#9271), total Akt (#9272), cytochrome c (#136F3), phospho-Erk 1/2 (Thr 202/Tyr 204) (#9101), total Erk 1/2 (#9102), poly/mono-ADP ribose (E6F6A) (#83732), phospho-signal transducer and activator of transcription 3 (STAT3, Tyr 705) (#9145), total STAT3 (#9139), phospho-Src family kinases (Tyr 416) (#2101), total c-Src (#2109), phospho-SAPK/JNK 1/2 (Thr 183/Tyr 185) (#4668), and GAPDH (#2118S) and HRP-conjugated anti-rabbit secondary antibodies were purchased from Cell Signaling Technology (Tokyo, Japan). The primary antibody against ATP5H (sc-515915) was purchased from Santa Cruz Biotechnology (Dallas, TX, USA).

### 2.2. Cell Culture

Two human GBM cell lines were used. T98G cells were obtained from Cell Lines Service (#300365) and U343MG cells from the American Type Culture Collection (#CRL-1690). The cells were cultured in a 10 cm dish containing RPMI 1640 medium supplemented with heat-inactivated 10% FBS and 1% penicillin–streptomycin and maintained in a humidified chamber as an adherent monolayer by serial passage at 37 °C under 5% CO_2_ conditions.

### 2.3. Small Interfering RNA (siRNA)-Mediated Gene Knockdown

siRNA-mediated gene knockdown was performed by transfection of 20 nmol·L^−1^ of Stealth Select RNAi (Thermo Fisher Scientific) using Lipofectamine RNAiMAX. Stealth RNAi™ siRNA Negative Control GC Duplex #3 (Thermo Fisher Scientific) was used as the negative control siRNA. The siRNAs targeting Bax or Bak were as follows.

Bax #1 (sense): 5′-ACUUUGCCAGCAAACUGGUGCUCAA-3′

Bax #2 (sense): 5′-GGCUGUUGGGCUGGAUCCAAGACCA-3′

Bak #1 (sense): 5′-AACAGAACCACACCCAGAACCACCA-3′

Bak #2 (sense): 5′-UGAAGAAUCUUCGUACCACAAACUG-3′

### 2.4. In Vitro NPe6-Mediated PDT Model of GBM Cells

NPe6 was dissolved in high-purity water at a concentration of 10 mg·mL^−1^ as a stock solution. Unless otherwise indicated, for NPe6-PDT, GBM cells were washed twice with phosphate-buffered saline (PBS) and then incubated with 0–20 μg·mL^−1^ NPe6 in a growth medium for 12 h. The cells were then irradiated with a 664 nm semiconductor laser beam (Panasonic Healthcare Co., Ltd. Osaka, Japan) at a radiation power density of 8.73 mW·cm^−2^ and total radiation energy density of 1.05 J·cm^−2^ and were continuously cultured for 1–14 days. Simultaneously, the time of laser irradiation was set as T_0_ for each assay, and various pharmacological inhibitors were added 2 h before laser irradiation according to the experiments (Appendix A).

### 2.5. Cell Toxicity Assay

At the indicated time points, the apoptosis rate of each treated cell was assayed by co-staining with PI and Ho, as previously described [15], with slight modifications. Fluorescence images of the stained cells were acquired under a CKX53 microscope (Olympus, Tokyo, Japan). At least 4000 cells per sample were counted using ImageJ software (NIH), and the ratio of PI-positive cells to total Ho-positive cells was calculated. 

### 2.6. Quantitation of Caspase-Dependent Apoptosis-To-Caspase-Independent Apoptosis Ratio

The apoptosis rate of cells following NPe6-PDT with or without pretreatment using the pan-caspase inhibitor z-VAD-FMK (z-VAD) was calculated. After 24 h, the mortality rate of each treated cell was quantified as described above, and the rate of caspase-dependent or -independent apoptosis was calculated as follows:Caspase-dependent apoptosis rate = apoptosis rate with z-VAD-FMK/(apoptosis rate with z-VAD-FMK + apoptosis rate without z-VAD-FMK)
Caspase-independent apoptosis rate = apoptosis rate without z-VAD-FMK/(apoptosis rate with z-VAD-FMK + apoptosis rate without z-VAD-FMK).

### 2.7. 5-FU Treatment of GBM Cells

As the positive control of apoptosis induction upon GBM cells, GBM cells were treated by 5-FU (500 μM) in the presence or absence of pretreatment by z-VAD (100 μM) for 2 h. After 24 h, the mortality rate of each treated cell was quantified.

### 2.8. Immunoblotting, In Vitro Protein Crosslinking Assay, and Cell Fractionation

Cell lysates were prepared and immunoblotting was performed as described previously [16]. The antibodies were diluted in Signal Enhancer HIKARI. Band signals were visualized with Western Lightning Plus-ECL and captured using Amersham Imager 600 (GE Healthcare, Chicago, IL, USA). Immunoblotting combined with in vitro protein linking was performed as previously described [17] with minor modifications. In brief, after washing twice with ice-cold PBS, the collected cells were lysed in CHAPS buffer (10 mM HEPES (pH 7.4), 150 mM NaCl, and 1% CHAPS) containing protease inhibitor cocktail (1/100) and phosphatase inhibitor cocktail (1/100) for 1 h on ice. Then, the protein crosslinker bismaleimidohexane (BMH) was added to the obtained cell lysates at a final concentration of 5 mM. After incubation for 30 min at 25 °C, the lysates were centrifuged at 15,000× *g* at 4 °C for 10 min. The supernatants were collected and analyzed by immunoblotting using a primary antibody (Bax) at a protein dose of 100 μg·sample^−1^. Cell fractionation was performed using the Cell Fractionation Kit (#ab109719; Abcam) according to the manufacturer’s protocol.

### 2.9. Measurement of Poly ADP-Ribose (PARP) Activity

PARP activity was measured by quantitation of the results of immunoblot analysis using anti-poly/mono-ADP ribose antibody and anti-GAPDH antibody. Quantitation of the results of immunoblotting was performed using ImageJ software, and PARP activity was calculated as follow: (poly/mono-ADP ribose bands density)/(GAPDH band density).

### 2.10. Wound Healing Assay

Wound healing assays were performed using the IncuCyte Zoom 96-well scratch wound cell migration assay protocol, according to the supplier’s instructions. Briefly, cells were seeded in 96-well ImageLock plates (#4379; Sartorius, Tokyo, Japan) at 30,000 cells·well^−1^ and incubated overnight. Subsequently, WoundMaker (#4493; Sartorius, Tokyo, Japan) was used to create uniform scratches in each well. After replacing the medium twice, the plates were placed on the IncuCyte Zoom (Sartorius, Tokyo, Japan) and a live image was recorded every 1 h for 24 h using a 10× objective. The live images were analyzed with IncuCyte to calculate relative wound density [RDW; a measure (%) of the density of the wound region relative to the density of the cell region].

### 2.11. Transwell Migration and Invasion Assay

Transwell migration and invasion assays were performed using 8-μm polycarbonate transwell filter chambers (#353097; Corning, NY, USA), as described previously [15]. For the invasion assay, the top surface of the transwell membranes was coated with Cellmatrix Type I-A. Cells were seeded at 25,000 cells·well^−1^ into the top chambers of 24-well transwells. A medium containing 10% FBS as a chemoattractant was placed in the bottom chamber of each transwell. After 24 h, residual or non-migrating/invading cells at the top surface of the transwell membranes were removed, and the membranes were fixed using 4% paraformaldehyde phosphate buffer solution. After fixation, the transwell membranes were stained with Ho, and images covering the bottom surface area of each membrane were recorded with a fluorescence microscope using a 40× objective. Cells in each image were counted, and the results were corrected by the number of cells when the treated cells were simply cultured in a 24-well plate. Migration assay was performed by the same procedure as invasion assay using the transwell without Cellmatrix Type I-A coating. 

### 2.12. Invadopodia Assay

Fluorescent matrix-coated coverslips were prepared as previously described [18]. A total of 50,000 cells were seeded on coverslips for 12 h. The cells were co-stained with Alexa 546 phalloidin and DAPI, and gelatin was labeled with FITC. To measure the gelatin-degradation activity of invadopodia, the degradation area observed in images was calculated using ImageJ 1.41 and the measurements were normalized to the total number of cells in each image. In each experiment, 20–21 randomly selected fields were recorded with a 40× objective and analyzed.

### 2.13. Statistical Analysis

All quantitative results, except those of invadopodia assay, are presented as the means ± standard deviations of three independent experiments and were analyzed by unpaired Student’s *t*-test using Microsoft Excel. One-way ANOVA and Tukey’s multiple comparison test were used to analyze the degradation area in invadopodia assay. The *p* values < 0.01 were considered significant.

## 3. Results

### 3.1. NPe6-PDT Induced Caspase-Dependent/Independent GBM Cell Death In Vitro through the Mitochondrial Pathway

First, to investigate the characteristics of GBM cells that survived NPe6-PDT (NPe6-PDT-R cells), we used the in vitro NPe6-PDT model as described above against several human GBM cell lines including patient-derived GBM cells in various experimental conditions. Among these GBM cells, only T98G and U343MG cells demonstrated repopulation within 14 days after in vitro NPe6-PDT that induced moderate cell death, resulting in acquisition of NPe6-PDT-R cells. Therefore, we applied these two GBM cells as the GBM model in this study. As shown in Figure 1a, the cell death of both T98G and U343MG cell lines, detected by the dye exclusion assay, was triggered by NPe6-PDT in an NPe6 dose-dependent manner. Moderate GBM cell death induced by PDT with 15 µg·mL^−1^ NPe6 in both lines was partially blocked by the pan-caspase inhibitor z-VAD, while moderate cell death induced by 5-FU, an anticancer agent that triggers GBM cell apoptosis [19], was almost completely inhibited by co-treatment with z-VAD (Figure 1b). Immunoblotting revealed that the cleavage of PARP, a substrate for effector caspases well-recognized as an apoptosis marker, was increased in cells subjected to NPe6-PDT and 5-FU treatment. Treatment with z-VAD completely inhibited PARP cleavage in cells subjected to 5-FU treatment as well as in those subjected to NPe6-PDT (Figure 1b). In addition, the rate of z-VAD-mediated inhibition of NPe6-PDT-induced GBM cell death decreased with increasing NPe6 doses (Figure 1c). These results suggest that NPe6-PDT triggers apoptotic cell death at a low rate (at a low NPe6 dose). More importantly, caspase-independent cell death occurs simultaneously with caspase-dependent cell death, and its rate increases with increasing NPe6 doses.

Based on this evidence, we next investigated whether Bax and Bak, the master regulators of mitochondrial caspase-dependent/independent cell death [20,21], are involved in NPe6-PDT-induced GBM cell death. To investigate this, lysates of T98G and U343MG cells subjected to NPe6 treatment, laser treatment, or NPe6-PDT without or with in vitro crosslinking using BMH were analyzed by immunoblotting to monitor Bax and Bak self-oligomerization. Bax and Bak self-multimerize to form pores on the mitochondrial outer membrane, thereby releasing cell death-inducing molecules from the mitochondria into the cytosol during mitochondria-dependent caspase-dependent/independent cell death [22], which is the most common programmed cell death pathway. Formation of Bax homodimers, the hallmark of mitochondrial caspase-dependent/independent cell death cascade activation, was confirmed only in cells subjected to NPe6-PDT (Figure 1d), suggesting the involvement of mitochondrion-dependent cell death signaling in NPe6-PDT-induced GBM cell death.

### 3.2. AIF and Poly (ADP-Ribose) Polymerase 1 (PARP1) Are Involved in NPe6-PDT-Induced Caspase-Independent GBM Cell Death

To explore the mechanisms underlying NPe6-PDT-induced caspase-independent mitochondria-dependent GBM cell death, the roles of AIF and PARP1, both of which regulate caspase-independent mitochondria-dependent cell death signaling upon various stimuli [23,24,25], were investigated. Immunoblotting analysis of the lysates of T98G and U343MG cells subjected to NPe6 treatment, laser treatment, or NPe6-PDT revealed marked poly ADP-ribosylation of intracellular proteins only following NPe6-PDT (Figure 2a), suggesting that NPe6-PDT mediated PARP1 activation and poly ADP-ribosylation of intracellular proteins in GBM cells.

Based on this evidence, we next explored whether PARP1 activity is essential for caspase-independent GBM cell death induced by NPe6-PDT. T98G and U343MG cells were subjected to NPe6-PDT in the presence or absence of z-VAD and/or a PARP inhibitor (3-aminobenzamide (3-ABA) or PJ34), as shown in Figure 2b, and the rate of cell death was quantified by the dye exclusion assay. Both 3-ABA and PJ34 suppressed NPe6-PDT-induced GBM cell death even in the presence of z-VAD (Figure 2b). Therefore, PARP1 is essential for the regulation of NPe6-PDT-induced caspase-independent cell death.

Generally, in PARP1-mediated mitochondria-dependent cell death, AIF, a mitochondrial flavoprotein, normally acts as nicotinamide adenine dinucleotide oxidase, which is released from the mitochondria into the cytosol, shuttles into the nucleus and induces caspase-independent cell death upon PARP1 activation. Therefore, the involvement of AIF in NPe6-PDT-induced GBM cell death was investigated. T98G and U343MG cells were subjected to NPe6-PDT (NPe6, 15 µg·mL^−1^) in the presence or absence of 3-ABA or PJ-34 (Figure 2c), and cell lysates were fractionated and assayed using immunoblotting. Cytosolic or nuclear translocation of AIF induced by NPe6-PDT was almost completely inhibited by 3-ABA or PJ-34 treatment (Figure 2c). Meanwhile, NPe6-PDT-triggered cytosolic release of cytochrome c, a mediator of mitochondria-dependent apoptosis released from the mitochondria into cytosol in response to apoptotic stimuli, was partially blocked by 3-ABA or PJ-34 treatment (Figure 2c). Simultaneous Bax and Bak knockdown also markedly reduced the cytosolic release of AIF and cytochrome c, as well as NPe6-PDT-induced T98G and U343MG cell death (Appendix A and Figure 2d), suggesting a critical role of the PARP1-mitochondrion-AIF axis in NPe6-PDT-induced caspase-independent cell death.

### 3.3. The Established In Vitro GBM Cell Model Survived NPe6-PDT

Upon confirming the mechanism of NPe6-PDT-induced GBM cell death, we next established NPe6-PDT-R cells by the continuous culture of T98G and U343MG cells following NPe6-PDT. As demonstrated in Figure 1a, PDT with 15 μg·mL^−1^ NPe6 triggered moderate-to-severe apoptosis in both T98G and U343MG cells 24 h after laser irradiation. Therefore, this NPe6 dose was used to establish NPe6-PDT-R cells. Within several days after NPe6-PDT, as described above, apoptosis of T98G and U343MG cells was induced, and the proliferation of both cells was halted (data not shown). However, 14 days after NPe6-PDT, both cells regrew, demonstrating resistance against NPe6-PDT-induced cell death compared with the control GBM cells initially subjected to NPe6-PDT (Figure 3a). Hence, we speculated that these GBM cells (NPe6-PDT-R cells) would mimic the clinical appearance of GBM relapsed after NPe6-PDT. At this time, NPe6-PDT-induced Bax dimerization and caspase-3 activation were also suppressed in NPe6-PDT-R cells compared with those in control cells (Figure 3a); therefore, resistance against NPe6-PDT-induced apoptosis may be regulated at the level upstream of the mitochondria.

### 3.4. Upregulated ERK1/2 Activation Is Essential for the Resistance of Surviving GBM Cells and GBM Stem Cells to NPe6-PDT-Induced Cell Death

Using NPe6-PDT-R T98G and U343MG cells in addition to control GBM cells, as mentioned above, the intracellular signaling essential for the maintenance of resistance against NPe6-PDT-induced cell death was investigated. Because the above results indicated that resistance against NPe6-PDT-induced cell death would be regulated upstream of the mitochondria, we explored the molecular signaling pathways occurring downstream of RTKs and stress-activated protein kinase/Jun amino terminal kinase (JNK), which play fundamental regulatory roles in cell survival under various stresses at the level upstream of most intracellular signaling pathways. By comparing the activation of molecules downstream of RTKs [Akt, STAT3, ERK1/2, and Src family kinases (SFKs)] as well as JNK1/2 by immunoblotting, we demonstrated that only ERK1/2 activation was upregulated in both NPe6-PDT-R cells (Figure 3b). Therefore, we hypothesized that sustained ERK1/2 activation in NPe6-PDT-R cells regulates the resistance to NPe6-PDT. To test this hypothesis, we additionally treated NPe6-PDT-R cells with trametinib, a clinically used inhibitor of MEK1/2 (upstream activator kinases of ERK1/2), simultaneously with NPe6-PDT and investigated whether trametinib reversed the resistance of GBM cells to NPe6-PDT. Treatment of NPe6-PDT-R cells with trametinib suppressed ERK1/2 activation and upregulated NPe6-PDT-induced Bax dimerization, thereby re-sensitizing cells to NPe6-PDT-induced cell death in a concentration-dependent manner (Appendix A, Figure 3c,d). However, trametinib did not affect the NPe6-PDT-induced cell death of control cells (Figure 3d). These results indicate the possible regulatory role of enhanced ERK1/2 activation in the resistance to NPe6-PDT-induced cell death via the suppression of Bax oligomerization. Further, we investigated the role of ERK1/2 in NPe6-survived GBM stem cells (GSCs); because GSCs are known to play the essential roles in the resistance of GBMs against various treatments. The GCSs of T98G cells (T98G-GSCs) were obtained by continuous culture of original T98G cells under serum-free growth factor-supplemented medium (see Appendix A and Methods) (Appendix A), and NPe6-PDT-R T98G-GSCs were established by continuous culture of T98G-GSCs following NPe6-PDT. As a result, NPe6-PDT-R T98G-GSCs were also resistant to NPe6-PDT compared with the control T98G-GSCs initially subjected to NPe6-PDT, and additional treatment of NPe6-PDT-R T98G-GSCs by trametinib reversed the resistance against NPe6-PDT (Appendix A). Collectively, the crucial roles of ERK1/2 in resistance against NPe6-induced cell death of NPe6-PDT-R GSC cells is also suggested. 

### 3.5. NPe6-PDT-R Cell Migration Is Enhanced via ERK1/2 Activation-Dependent Machinery

Previous in vitro evidence suggested that GBM cells with acquired resistance to temozolomide, an alkylating agent used in current standard GBM therapy, demonstrate higher invasiveness, and relapsed GBM cases following treatment often exhibit a higher infiltrative phenotype. In addition to these previous findings, distorted morphology of T98G and U343MG cells 5 days after NPe6-PDT, compared with that of the control cells, was confirmed in this study (Appendix A). Hence, we hypothesized that the GBM relapsed after NPe6-PDT acquires a more invasive phenotype and tested whether the migration or invasion abilities of NPe6-PDT-R cells were upregulated compared with those of the control cells. First, the migration abilities of control and NPe6-PDT-R cells were investigated using the transwell assay during 14 days after NPe6-PDT. The migration abilities of NPe6-PDT-R T98G and U343MG cells were upregulated, particularly at 5 days after NPe6-PDT, compared with those of the control cells (Figure 4a). Therefore, the molecular mechanisms underlying the upregulated migration abilities of both cells, specifically those related to RTK-mediated growth signaling and the representative stress kinase JNK1/2, which are important for cell survival as well as cell migration/invasion, were further investigated by immunoblotting. As shown in Figure 4b, only ERK1/2 phosphorylation (=activation) was correlated to the migration ability of the control and NPe6-PDT-R cells, which was also demonstrated to regulate the resistance of cells against NPe6-PDT-induced cell death (Appendix A, Figure 3b–d); therefore, we further explored the effects of inhibition of ERK activity on the enhanced migration ability of NPe6-PDT-R cells. The transwell assay demonstrated that the enhancement of the migration abilities of NPe6-PDT-R T98G and U343MG cells at 5 days after NPe6-PDT was suppressed by concomitant treatment with trametinib or U0126, another MEK1/2 inhibitor, which also effectively suppressed ERK1/2 activation, as observed in the immunoblotting analysis (Figure 4d). In addition, wound healing assay using the Incucyte^®^ system revealed that the migration abilities of NPe6-PDT-R T98G and U343MG cells were also enhanced compared with those of the control cells, which was suppressed by co-treatment with trametinib (Figure 4d). Taken together, these results indicate the essential roles of ERK1/2 in the positive regulation of the migration ability of NPe6-PDT-R cells.

### 3.6. Degradation of Extracellular Matrix (ECM) Is Promoted in NPe6-PDT-R Cells Independent of ERK Activation

Because in addition to cellular migration capacity, adjacent ECM degradation is also recognized as a fundamental functional component for undergoing cellular invasion, invadopodia formation, a hallmark of cell invasion-triggered ECM degradation, was explored in U343MG NPe6-PDT-R cells to further investigate GBM cell invasiveness following NPe6-PDT. At 3 days, U343MG cells subjected to NPe6-PDT exhibited invadopodia formation, as opposed to the control, NPe6-treated, or laser-treated cells (Figure 5a). However, co-treatment with trametinib did not suppress invadopodia formation in cells subjected to NPe6-PDT (Figure 5b). Essentially the same results were obtained in the same assay at 5 days (data not shown). Thus, we further investigated the invasion of U343MG cells subjected to NPe6-PDT using the collagen-coated transwell invasion assay.

At 5 days after treatment, the invasion of T98G and U343MG cells subjected to NPe6-PDT was enhanced compared with that of control cells, and co-treatment with trametinib effectively blocked this enhanced invasion (Figure 5c). Therefore, ERK1/2 plays pivotal roles in the regulation of the elevated invasiveness of GBM cells surviving NPe6-PDT, independent of the regulation of enhanced ECM degradation.

## 4. Discussion

There are few reports regarding the molecular machinery regulating NPe6-PDT-induced cell death or resistance to NPe6-PDT. Among these mechanisms, mitochondrion-mediated intrinsic cell death pathway-related signaling has been reported to be specifically involved [26,27,28]. For instance, in a lung adenocarcinoma cell line, NPe6-PDT has been reported to trigger mitochondrial proapoptotic Bax activation-dependent cell death by inducing lysosomal damage [26]. In addition, in breast cancer cell lines, the expression of Bcl-2, a representative antiapoptotic Bcl-2 family protein, which associates with and inhibits the expression of Bax, Bak, and other proapoptotic Bcl-2 family proteins, thereby preventing intrinsic mitochondria-dependent caspase-dependent/independent cell death, was decreased following PDT with photofrin, another photosensitizer, or NPe6-PDT [28]. Furthermore, in breast cancer cells, the expression of survivin, an inhibitor of the expression of apoptotic proteins that suppresses mitochondrion-mediated cell death by inhibiting the activities of caspases and AIF downstream of the mitochondria, was upregulated following NPe6-PDT [28]. Accordingly, heat-shock protein 90 (HSP90), which stabilizes survivin, has been proposed as a potent target for enhancing the antitumor effects of NPe6-PDT [28]. Consistent with these previous reports, our results also confirmed the essential roles of mitochondrion-mediated caspase-dependent/independent cell death and its core regulators, including the proapoptotic Bcl-2 family proteins Bax and Bak (Figure 1, Figure 2 and Figure 3). In previous studies, the essential roles of reactive oxygen species (ROS) in regulation of PDT-induced caspase-dependent/independent cell death were shown [29,30], although the precise molecular machinery remains unknown. In the present study, caspase-dependent apoptosis and the detailed molecular mechanism underlying potent NPe6-PDT-induced caspase-independent GBM cell death via the PARP1-Bax/Bak-AIF axis was proposed. However, we did not determine whether ROS regulate PARP1 activation-mediated NPe6-PDT-induced GBM cell death, because antioxidants did not significantly block NPe6-PDT-induced GBM cell death in our model (data not shown). Therefore, the molecular mechanisms regulating PARP1 activation-mediated NPe6-PDT-induced GBM cell death in our model warrant further careful investigation.

Meanwhile, aberrant PARP1 activation also activates caspases via mitochondrial Bax activation [23]. Corroborating this, treatment with a PARP inhibitor also partially blocked cytochrome c release from the mitochondria into the cytosol in this study (Figure 2c); therefore, PARP1 might regulate both caspase-dependent/independent cell death via proapoptotic Bcl-2 family protein activation in NPe6-PDT-dependent GBM cell death. In the present study, we did not demonstrate whether AIF directly contributes to NPe6-PDT-induced caspase-independent cell death. Nonetheless, the crucial roles of mitochondrial Bax and Bak in NPe6-PDT-induced GBM cell death were proved, confirming that caspase-independent cell death-inducing factors including AIF, which are released from the mitochondria into the cytosol via mitochondrial Bcl-2-dependent pore formation triggered by cell death signal transduction, are also involved in NPe6-PDT-induced GBM cell death [31,32]. Consequently, based on our results, PARP1, proapoptotic Bcl-2 family proteins, and AIF may be potent regulators of NPe6-PDT-induced mitochondrion-dependent caspase-dependent/independent GBM cell death.

Although the subcellular localization of NPe6 during NPe6-PDT in GBM cells has not been explicitly clarified, previous report demonstrated that NPe6 is diffusely presented in the early post-dose period but exhibits lysosome localization with time in the cells and also suggested this alteration of intracellular NPe6 localization determines the cell death phenotype in rat myocytes [33]. In our study, flow cytometry measurements of NPe6 uptake following NPe6 administration in GBM cells showed that NPe6 uptake increased with time and that NPe6 was taken up by almost all GBM cells after 4 h of treatment, but specific subcellular localization could not be pursued thereafter (data not shown). Therefore, we considered NPe6 might show similar changes of subcellular localization in GBM cells thereby contribute to alteration of cell death phenotype in the present study.

As a crucial regulator of cell survival following PDT, sustained upregulation of ERK activity within a maximum of 12 h following PDT, which played an essential role in cell survival after NPe6-PDT in our study, has already been demonstrated previously [34,35]. In addition, kinetics of kinase-mediated stress response signaling following PDT have been investigated within no longer than 1 day after treatment [34,35,36,37]. Conversely, in the present study, we focused on the subacute or later (2 days onward) phase following NPe6-PDT of GBM cells and investigated the behaviors of kinases associated with cellular stress and survival. We observed transient (specifically within 2–6 days after PDT) and constitutive (14 days or more after PDT) ERK activation following NPe6-PDT (Figure 3b and Figure 4b). In addition, we demonstrated that pharmacological suppression of ERK activation at specific time points (days 3 and 14) after NPe6-PDT effectively inhibited the migration or invasion of NPe6-PDT-R GBM cells, whereas the pharmacological inhibition of ERK activation in NPe6-PDT-R cells at 14 days after PDT enhanced NPe6-PDT-induced NPe6-PDT-R cell death. Based on these results, at least in our model, sustained ERK1/2 activation during the subacute or later phase plays pivotal roles in the migration or invasion and survival of NPe6-PDT-R GBM cells. However, the detailed molecular mechanisms modulating ERK1/2 activation remain unknown. Under severe cellular stresses, both phosphorylation-mediated signal transduction and dephosphorylation-mediated negative feedback are triggered in RTK signaling [38,39,40,41,42,43]. In particular, in ERK1/2-dependent signaling, dephosphorylation of ERK1/2- and ERK1/2-dependent signaling molecules by protein phosphatases, such as protein phosphatase 2A (PP2A) or dual-specificity phosphatases (DUSPs), has previously been demonstrated [41,42,43]. Therefore, we investigated the expression of PP2A subunit C and DUSP6 by immunoblotting; however, the expression levels of both phosphatases in NPe6-DT-R cells were not altered during 14 days after NPe6-PDT (data not shown). In addition, both the activation and total expression levels of the kinases were drastically altered during the same period. Therefore, ERK1/2 activation may have been upregulated as a result of various stress responses simultaneously induced by potent NPe6-PDT.

Understanding and regulating the mechanisms underlying tumor cell invasiveness into the surrounding normal tissues have always been hurdles in the treatment of highly invasive malignant tumors, including GBMs. For invasion into adjacent tissues, the cells must migrate toward the neighboring normal tissues and degrade the ECM of adjacent cells; thus, targeting either or both of the machineries is considered crucial to prevent tumor cell invasion [44,45]. In the present study, suppression of the upregulated MAPK cascade inhibited the elevated migration capacity of GBM cells in the multiple migration assay but failed to block the augmented ECM degradation in NPe6-PDT-R cells, as evidenced by invadopodia assay. In addition, cell invasion assay using collagen-coated transwells revealed the suppression of the upregulated invasiveness of NPe6-PDT-R GBM cells. Therefore, the inhibition of ERK activation alone might be sufficient for blocking the upregulated invasiveness of NPe6-PDT-R GBM cells. Further investigation using similar in vivo models is warranted to confirm the potential of ERK activation as a therapeutic target for preventing the re-growth and invasion of NPe6-PDT-R GBM cells.

## 5. Conclusions

In the present study, by using GBM cells that regrew within a short period after NPe6-PDT, we demonstrated the detailed molecular mechanisms underlying NPe6-PDT-induced GBM cell death and revealed that NPe6-PDT likely induces a more malignant phenotype of GBM following relapse through an ERK activation-dependent machinery. Since NPe6-PDT was relatively recently approved for the clinical management of GBMs, tumor resistance against this treatment is not a major problem. Nevertheless, similar to other available therapies for GBMs, resistance against—and relapse following—NPe6-PDT, giving rise to more malignant phenotypes, are considered critical problems in the near future. In this light, the signaling cascades or molecules elucidated in the present study may be promising candidate therapeutic targets for GBM.

## Figures and Tables

**Figure 1 cancers-12-03641-f001:**
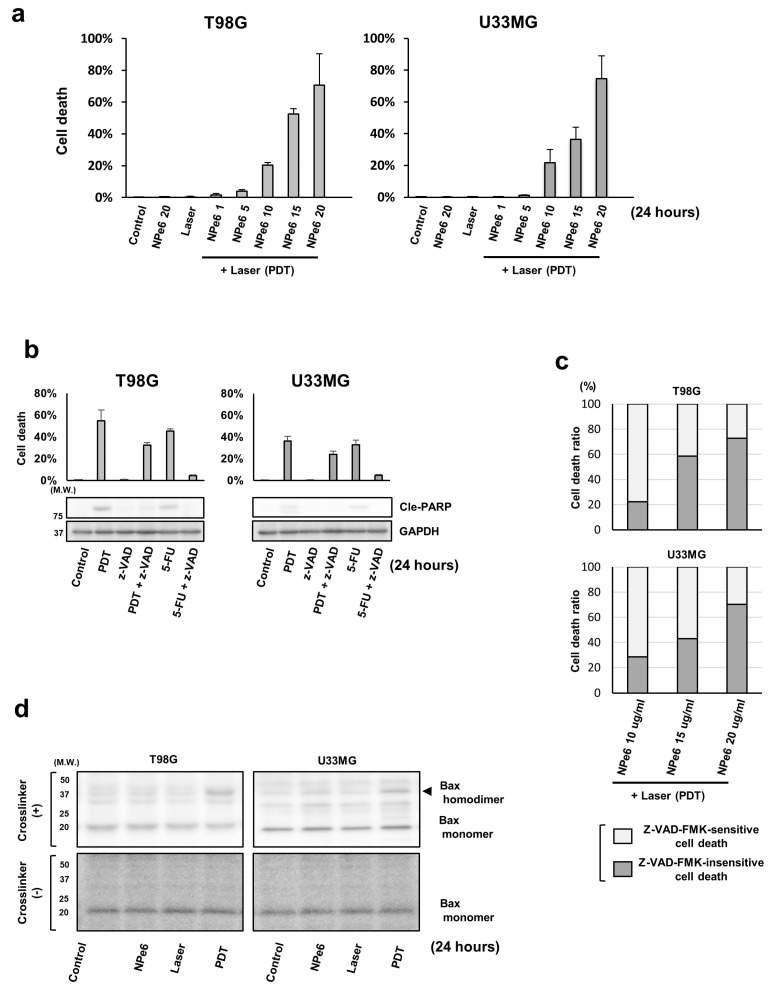
Talaporfin (NPe6)-mediated photodynamic therapy (NPe6-PDT) induced proapoptotic Bcl-2 associated X protein (Bax) activation and caspase-dependent/independent cell death in glioblastoma (GBM) cells. (**a**) Human GBM cell lines T98G and U343MG were subjected to vehicle treatment (control), NPe6 treatment (μg·mL^−1^), laser treatment, or NPe6-PDT, as shown in the figure (see the Materials and Methods for detailed treatments). After 24 h, the mortality rate of the treated cells was assayed by the dye exclusion assay. (**b**) T98G and U343MG cells were subjected to vehicle treatment (control), NPe6-PDT (NPe6, 15 μg mL^−1^), or 5-fluorouracil (5-FU, 500 μM) treatment with or without z-VAD-FMK (a pan-caspase inhibitor; 100 μM) pretreatment for 2 h before laser irradiation (Figure 1A), as shown in the figure. After 24 h of laser irradiation, the mortality rate was assayed as described in the Materials and Methods. (**c**) T98G and U343MG cells were subjected to NPe6-PDT at the indicated NPe6 dose with pretreatment using vehicle or z-VAD-FMK (100 μM), as shown in the figure. After 24 h, the relative proportion of the mortality rates of z-VAD-FMK-sensitive and -insensitive cells in total cell death rate were calculated as described in the Materials and Methods. (**d**) T98G and U343MG cells were subjected to vehicle treatment (control), NPe6 treatment (15 μg·mL^−1^), laser treatment, or NPe6-PDT (NPe6 15 μg·mL^−1^), as shown in the figure. After 24 h, the cell lysates were subjected to in vitro protein crosslinking assay as described in the Materials and Methods, followed by immunoblotting using a primary antibody against Bax.

**Figure 2 cancers-12-03641-f002:**
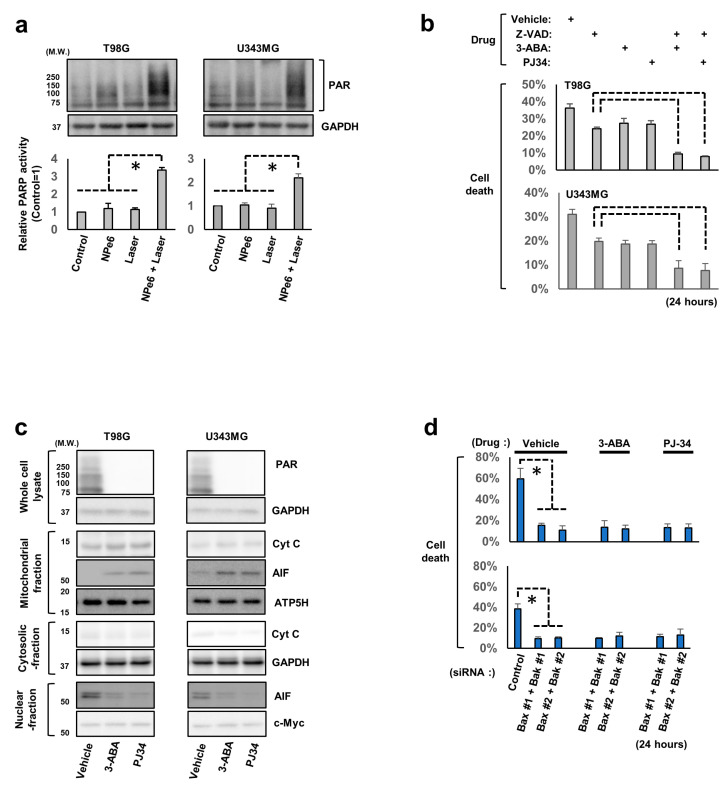
NPe6-PDT-induced cell death is mediated by the PARP1 activation-mediated activation of the mitochondrial proapoptotic Bcl-2 family proteins. (**a**) T98G and U343MG cells were treated as shown in Figure 1D. After 24 h, the lysates of these treated cells were analyzed by immunoblotting using the indicated primary antibodies (Upper). GAPDH antibodies were used to confirm the amounts of protein loaded. The relative PARP activity of the equally treated cells (control = 1) was also demonstrated (Lower, * *p* < 0.01). (**b**) T98G and U343MG cells were subjected to NPe6-PDT (NPe6, 15 μg·mL^−1^) in the presence or absence of pretreatment with the vehicle, z-VAD-FMK (100 μM), 3-aminobenzamide (3-ABA, a PARP inhibitor, 3 mM), or PJ34 (a PARP inhibitor, 15 μM), as shown in the figure. After 24 h, the mortality rate of all treated cells was determined. * *p* < 0.01. (**c**) T98G and U343MG cells were subjected to NPe6-PDT (NPe6, 15 μg·mL^−1^) in the presence or absence of pretreatment with vehicle, 3-ABA (3 mM), or PJ34 (15 μM), as shown in the figure. After 24 h, the lysates of all these treated cells were separated into the mitochondria-enriched membrane fraction, cytosolic fraction, and nuclear fraction, as described in the Materials and Methods. The fractionated cell lysates as well as whole-cell lysates were analyzed by immunoblotting using the indicated primary antibodies. Antibodies against GAPDH (cytosolic fraction marker), ATP5H (mitochondrial fraction marker), and c-Myc (nuclear fraction marker) were used to confirm the amount of proteins loaded in each fraction. (**d**) T98G and U343MG cells were treated with siRNAs as indicated. After 48 h, these cells were further subjected to NPe6-PDT (NPe6, 15 μg mL^−1^) in the presence or absence of pretreatment with vehicle, 3-ABA (3 mM), or PJ34 (15 μM), as shown in the figure. After 24 h, the cell death rate of all treated cells was calculated. * *p* < 0.01.

**Figure 3 cancers-12-03641-f003:**
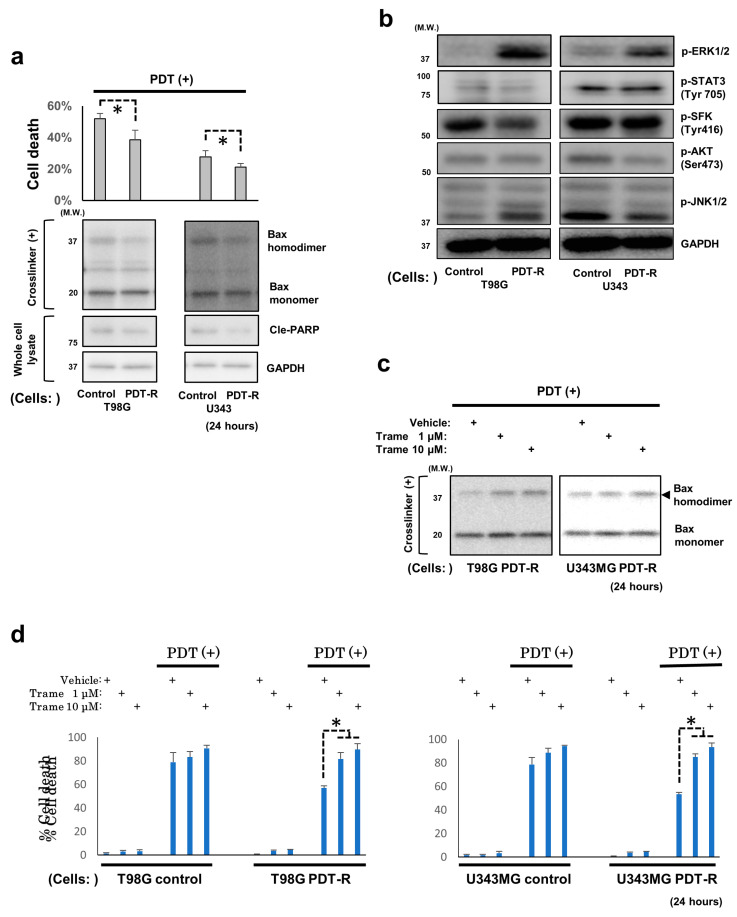
Enhanced extracellular regulated kinase1/2 (ERK1/2) activation positively regulates the resistance of GBM (NPe6-PDT-R) cells to NPe6-PDT-induced cell death via the suppression of Bax activation. (**a**) Control, as well as NPe6-PDT-R T98G and U343MG cells were subjected to NPe6-PDT (NPe6, 15 μg·mL^−1^). After 24 h, the mortality rate of these treated cells was calculated (upper, **p* < 0.01). Whole-cell lysates of control cells and equally treated cells subjected to *in vitro* crosslinking assay were analyzed by immunoblotting using the indicated primary antibodies (lower). (**b**) Lysates of control cells as well as those of NPe6-PDT-R T98G and U343MG cells were analyzed by immunoblotting using the indicated primary antibodies. (**c**) NPe6-PDT-R T98G and U343MG cells were subjected to NPe6-PDT (NPe6, 15 μg·mL^−1^), along with treatment with the vehicle or the indicated dose of trametinib. After 24 h, the lysates of crosslinked treated cells were analyzed by immunoblotting using a primary antibody against Bax. (**d**) Control cells, as well as NPe6-PDT-R T98G and U343MG cells were subjected to NPe6-PDT (NPe6, 15 μg·mL^−1^) in the presence or absence of pretreatment with the vehicle or the indicated dose of trametinib, as shown in the figure. After 24 h, the cell death rate of all treated cells was determined. * *p* < 0.01.

**Figure 4 cancers-12-03641-f004:**
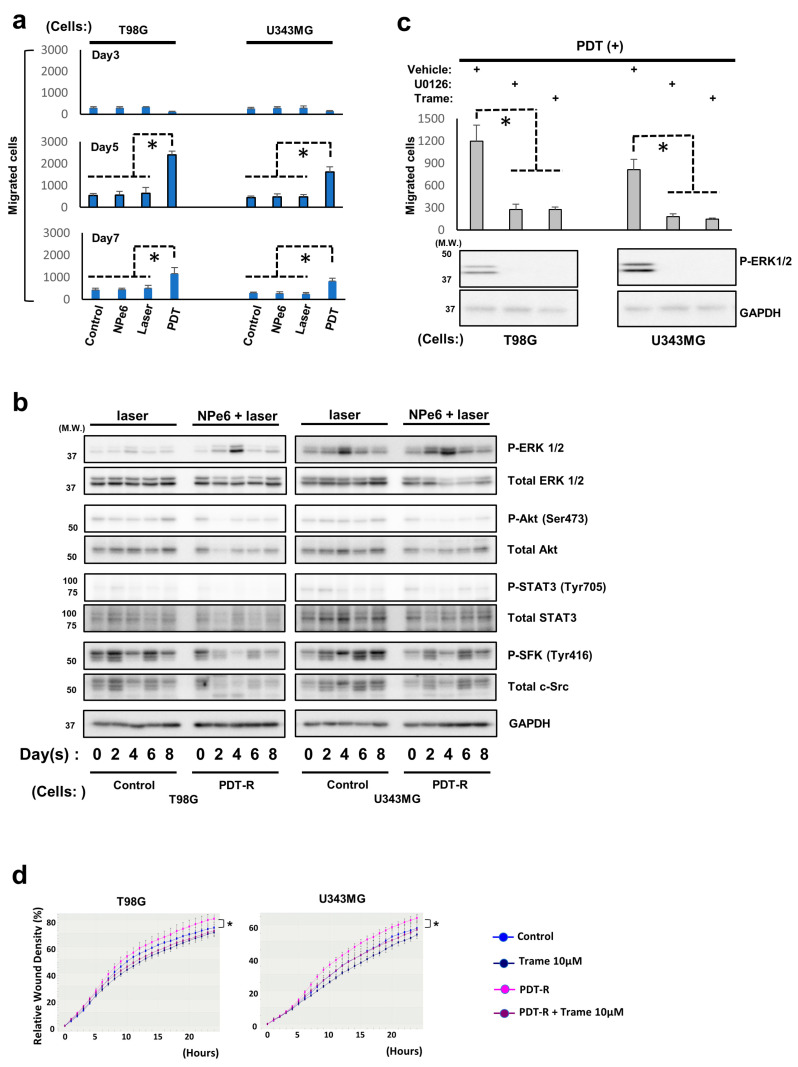
Enhanced migration of NPe6-PDT-R cells is regulated by enhanced ERK1/2 activation. (**A**) Control and NPe6-PDT-R T98G and U343MG cells were photographed under a phase-contrast microscope (×400). Scale bar, 10 μm. (**a**) T98G and U343MG cells were subjected to vehicle treatment (control), NPe6 treatment (15 μg·mL^−1^), laser treatment, or NPe6-PDT (NPe6, 15 μg·mL^−1^), as shown in the figure. Migration of these treated cells was assayed by the transwell assay at the indicated time points. * *p* < 0.01. (**b**) T98G and U343MG cells were subjected to laser treatment or NPe6-PDT (NPe6, 15 μg·mL^−1^) as shown in the figure. Then, the lysates of all treated cells at the indicated time points after treatment were analyzed by immunoblotting using the indicated antibodies. (**c**) NPe6-PDT-R T98G and U343MG cells were treated with vehicle, U0126 (MEK1/2 inhibitor; 10 mM), or trametinib (10 μM), as shown in the figure. After 24 h, the migration of all treated cells was assayed by the transwell assay (upper, * *p* < 0.01), and the lysates of the equally treated cells were analyzed by immunoblotting using the indicated primary antibodies (lower). (**d**) NPe6-PDT-R T98G and U343MG cells were treated as shown in Figure 4d. The migration abilities of all treated cells were analyzed using the Incucyte system, as described in the Materials and Methods. * *p* < 0.01.

**Figure 5 cancers-12-03641-f005:**
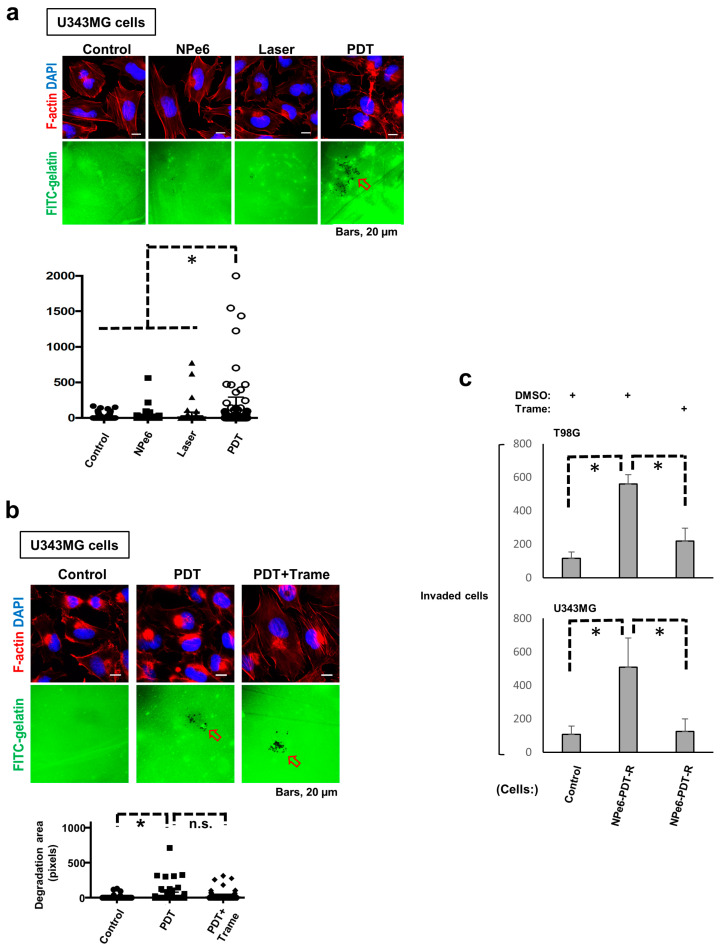
Invasion of NPe6-PDT-R cells is augmented in an ERK1/2 activation-dependent manner, but extracellular matrix degradation in NPe6-PDT-R cells is upregulated independent of ERK1/2 activation. (**a**) U343MG cells were subjected to vehicle treatment, NPe6 treatment (15 μg·mL^−1^), laser treatment, or NPe6-PDT (NPe6, 15 μg·mL^−1^) as shown in the figure. After 3 days, these cells were subjected to the invadopodia assay as described in the Materials and Methods. Representative fluorescence images of confocal microscopy (upper) [red fluorescence, phalloidin (actin); blue fluorescence, DAPI (nucleus); green fluorescence, FITC-gelatin] and invadopodia formation (lower, * *p* < 0.01) were examined. Arrowhead denotes degradation sites on the gelatin matrix. (**b**) U343MG cells were subjected to vehicle treatment (control) and/or NPe6-PDT (NPe6, 15 μg·mL^−1^). After 3 days, the cells were subjected to invadopodia assay with or without trametinib pretreatment (10 μM) for 12 h, and the results are shown in Figure 5A. Arrowheads denote degradation sites on the gelatin matrix. * *p* < 0.01. (**c**) T98G and U343MG cells were subjected to vehicle treatment (control) or NPe6-PDT (NPe6-PDT-R, NPe6, 15 μg·mL^−1^). After 5 days, the cells were subjected to transwell invasion assay with vehicle, trametinib (DMSO), or trametinib (10 μM) pretreatment for 12 h, as described in the Materials and Methods. * *p* < 0.01.

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
