# Peer review of "Enhanced Malignant Phenotypes of Glioblastoma Cells Surviving NPe6-Mediated Photodynamic Therapy are Regulated via ERK1/2 Activation"

_cancers, 2020, doi:10.3390/cancers12123641_

Round 1

Reviewer 1 Report

Could a comment be made (in conclusion) about the utility/ appropriateness of the 2 cell lines used in this work - particularly with regard to disease recurrence and the likely cell population that mediates this (ie glioma stem like cells)?

Are the 2 cell lines demonstrated here the most appropriate to demonstrate this effect?  Have the authors used any GSC cultures (human) to prove their findings further ?  

What is known in the literature about NPe6-PDT in GSC in vitro culture ?

Author Response

# Reviewer1

  1. Could a comment be made (in conclusion) about the utility/ appropriateness of the 2 cell lines used in this work - particularly with regard to disease recurrence and the likely cell population that mediates this (ie glioma stem like cells)?

  1. Are the 2 cell lines demonstrated here the most appropriate to demonstrate this effect?  Have the authors used any GSC cultures (human) to prove their findings further ?  

⇒ Thank you for the incisive comments above (1 and 2).

In this study, we originally tested multiple cell lines including other commercially available human GBM cell lines or patient derived GBM cell lines under various experimental conditions of NPe6-PDT. However, only T98G and U343MG cells demonstrated repopulation within two weeks after in vitro NPe6-PDT that induced moderate cell death, resulting in acquisition of NPe6-PDT-R cells. Therefore, we finally applied these two cell lines for this study. We added the sentences described about this in the “Results” (line 244-248) and “Conclusion” (line 561).

In addition, we have also checked the effect of this study in glioma stem cells (GSCs) of T98G cells maintained by the serum-free growth factors-supplemented culture medium and obtained nearly the same results as the original T98G cells maintained by the culture medium containing serum. Therefore, we added the results of the experiments using T98G GSCs in the supplementary figure S4 and added the sentences described about this result in in the “Abstract” (line 56-58) and “Results” (line 390-399).

  1. What is known in the literature about NPe6-PDT in GSC in vitro culture?

⇒ Thank you for the comment. As far as we know, there are still no previous reports described about the effect of NPe6-PDT against GSC in vitro.

Reviewer 2 Report

The manuscript submitted by Kobayashi et al. deals with the problem of glioblastoma PDT and emergence of talaporfin PDT resistant cells. ERK1/2 activation has already been known to result in PDT resistance in cases of some other photosensitizers and cancer types. This issue is important from the standpoint of basic science as well as further studies on this subject might potentially have an impact on the decisions made by clinicians. Because of that I think that is fits well the scope of the Cancers journal very well.

The article is well written, the methodology is solid. I would recommend accepting the article after addressing some minor suggestions:

  1. Did you consider the localization of the photosensitizers in the cells? It may vary depending on the concentration of the photosensitizer used, which may have an impact on the resulting mechanism of PDT.
  2. The figures and figure captions seem very “dense”. There is a lot of information given in a limited space which makes is sometimes difficult to follow the data presented.
  3. The sentence in line 485-487: “In previous studies, PDT-induced caspase-dependent/independent cell death was considered to be triggered mainly by reactive oxygen species (ROS) [29, 30], although the precise molecular machinery remains to be elucidated.” – you might consider rephrasing it.
  4. In some places the language could be a bit polished. There are some typos to be corrected (like in line 82 – there should be “chlorin” instead of “chlorine”).

Author Response

# Reviewer2

The manuscript submitted by Kobayashi et al. deals with the problem of glioblastoma PDT and emergence of talaporfin PDT resistant cells. ERK1/2 activation has already been known to result in PDT resistance in cases of some other photosensitizers and cancer types. This issue is important from the standpoint of basic science as well as further studies on this subject might potentially have an impact on the decisions made by clinicians. Because of that I think that is fits well the scope of the Cancers journal very well.

The article is well written, the methodology is solid. I would recommend accepting the article after addressing some minor suggestions:

  1. Did you consider the localization of the photosensitizers in the cells? It may vary depending on the concentration of the photosensitizer used, which may have an impact on the resulting mechanism of PDT.

  ⇒  Thank you for the comments; it was very perspective. Although the subcellular localization of NPe6 during NPe6-PDT in GBM cells has not been explicitly clarified, previous report demonstrated that NPe6 is diffusely presented in the early post-dose period but exhibits lysosome localization with time in the cells and also suggested this alteration of intracellular NPe6 localization determines the cell death phenotype in rat myocytes. [1] In our study, flow cytometry measurements of NPe6 uptake following NPe6 administration in GBM cells showed that NPe6 uptake was increased with time and that NPe6 was taken up by almost all cells after 4 hours of treatment, but specific subcellular localization could not be pursued thereafter (data not shown). Therefore, we considered NPe6 might show similar changes of subcellular localization in GBM cells thereby contribute to alteration of cell death phenotype in the present study. We described about above in “Discussion” section (line 513-521) with a newly added reference (Ref. 33).

[Refference]

1 The mechanism of PDT-induced electrical blockade: The dependence of time-lapse localization of talaporfin sodium on the cell death phenotypes in rat cardiac myocytes

       2. The figures and figure captions seem very “dense”. There is a lot of information given in a limited space which makes is sometimes difficult to follow the data presented.

  ⇒ Thank you for the comments. Following your advice, we transferred some of the data in figure1-4 to the supplementary figures (Supplementary Figure S1-S3, and S5). And the shifted descriptions of each figure through whole text were corrected.  

     3. The sentence in line 485-487: “In previous studies, PDT-induced caspase-dependent/independent cell death was considered to be triggered mainly by reactive oxygen species (ROS) [29, 30], although the precise molecular machinery remains to be elucidated.” – you might consider rephrasing it.

⇒ Thank you for the comments. According to your suggestion, we changed the sentence you pointed out (line 491-493) as followed : “In previous studies, the essential roles of reactive oxygen species (ROS) in regulation of PDT-induced caspase-dependent/independent cell death were shown [29, 30], although the precise molecular machinery remains unknown”.  

  1. In some places the language could be a bit polished. There are some typos to be corrected (like in line 82 – there should be “chlorin” instead of “chlorine”).

  ⇒ Thank you so much for pointing it out, and we corrected the spelling that were pointed out (line 83).

Reviewer 3 Report

Very detailed and careful work. The authors have created a cell culture model that credibly simulates resistance of the NPe6-mediated photodynamic 3 therapy. They then looked for factors of resistance and identified a key factor and successfully tested an effective inhibitor.

The results are, in my opinion, very valuable and could prove very useful in clinical use over the next few years.

Only the legibility of the work is sometimes a bit tedious, but this is probably primarily due to the complexity of the content.

Perhaps a sentence could be inserted into the abstract explaining why which proteins were selected for immunoblotting in line 42 for example "Immunoblotting for key proteins of candidate pathways" or "proteins that are known to be stress-induced" or similar.

This is explained very well in the discussion.

Author Response

# Reviewer3

Very detailed and careful work. The authors have created a cell culture model that credibly simulates resistance of the NPe6-mediated photodynamic 3 therapy. They then looked for factors of resistance and identified a key factor and successfully tested an effective inhibitor.

The results are, in my opinion, very valuable and could prove very useful in clinical use over the next few years.

Only the legibility of the work is sometimes a bit tedious, but this is probably primarily due to the complexity of the content.

Perhaps a sentence could be inserted into the abstract explaining why which proteins were selected for immunoblotting in line 42 for example "Immunoblotting for key proteins of candidate pathways" or "proteins that are known to be stress-induced" or similar.

This is explained very well in the discussion.

⇒ Thank you very much for the comments. As suggested above, we modified the "Abstract" (line 62).